# Urban–Rural Disparities in Community Participation after Spinal Cord Injury in Ontario

**DOI:** 10.3390/healthcare12202089

**Published:** 2024-10-20

**Authors:** Mohammadreza Amiri, Mohammad Alavinia, Farnoosh Farahani, Natavan Khasiyeva, Meredith Burley, Suban Kangatharan, Beverley Catharine Craven

**Affiliations:** 1Centre for the Business and Economics of Health, The University of Queensland, St. Lucia, QLD 4072, Australia; m.amiri@uq.edu.au; 2KITE Research Institute, University Health Network, Toronto, ON M5G 2A2, Canada; mohammad.alavinia@uhn.ca (M.A.); farnoosh.farahani@uhn.ca (F.F.); suban.kangatharan@uhn.ca (S.K.); 3Spinal Cord Injury Ontario, Toronto, ON M4G 3V9, Canada; navi.khasiyeva@sciontario.org (N.K.); meredith.burley@sciontario.org (M.B.); 4Division of Physical Medicine and Rehabilitation, Department of Medicine, Temerty Faculty of Medicine, University of Toronto, Toronto, ON M5S 3H2, Canada; 5Institute of Health Policy Management and Evaluation, University of Toronto, Toronto, ON M5S 1B2, Canada

**Keywords:** spinal cord injury or disease (SCI/D), community participation, Moorong Self-Efficacy Scale (MSES), Reintegration to Normal Living Index (RNLI), artificial intelligence, optical mark recognition (OMR)

## Abstract

Background: Personal, social, and environmental factors may influence self-efficacy and social reintegration among people living with spinal cord injury or disease (SCI/D) in urban and rural areas. Novel data collection methods have the potential to characterize community participation (CP) in diverse settings. Objectives: The objectives were (1) to describe and compare self-reported community participation (Reintegration to Normal Living Index (RNLI) and Moorong Self-Efficacy Scale (MSES)) levels of individuals with SCI/D living in urban or rural Ontario, Canada; and (2) to determine the accuracy of an artificial intelligence (AI) optical mark recognition tool for extracting data from CP surveys conducted among participants after transitioning from inpatient rehabilitation to home and residing in the community. Methods: We partnered with SCI Ontario staff to collect MSES and RNLI survey data from adults with motor complete (e.g., AIS A–B) and incomplete (AIS C–D) SCI/D living in urban or rural Ontario, Canada, between January and October 2022. The Rurality Index of Ontario (RIO) from the postal code determined urban or rural residency. Optical mark recognition (OMR) software was used for survey data extraction. A Research Associate validated the extracted survey responses. Descriptive statistics, correlation analysis, and non-parametric statistics were used to describe the participants, their impairments, and their reported CP levels across urban and rural settings. Results: Eighty-five individuals with SCI/D (mean age 53.7 years, 36.5% female) completed the survey. Most of the participants resided in major urban areas (69.4%) and had traumatic injuries (64.7%). The mean total MSES score for Ontarians with SCI/D was 87.96 (95% confidence interval [CI]: 84.45, 91.47), while the mean total RNLI score for the same individuals was 75.61 (95% CI: 71.85, 79.37). Among the MSES domains, the lowest score was observed in response to sexual satisfaction (mean: 4.012, 95% CI: 3.527, 4.497), while the lowest RNLI domain item score was associated with the ability to travel out of town (mean: 5.965, 95% CI: 5.252, 6.678). Individuals with incomplete injuries in rural areas reported lower MSES and RNLI scores than those with complete motor injuries, whereas no significant differences were found in MSES and RNLI scores among urban residents based on impairment. These findings suggest that, depending on the environmental context (e.g., rural vs. urban areas), AIS categories may influence the perception of CP among people living with SCI/D. The OMR tool had 97.4% accuracy in extracting data from the surveys. Conclusions: The CP (MSES and RNLI) scores reported by individuals with SCI/D differ based on their living setting. In rural Ontario, individuals with greater functional ability reported lower CP than their counterparts living in urban settings. Although CP remains a challenge, the needs of individuals with motor incomplete SCI/D and heterogeneous levels of mobility residing in rural areas require exploration and targeted interventions. The OMR tool facilitates accurate data extraction from surveys across settings.

## 1. Introduction

Spinal cord injury (SCI) or disease (SCI/D) results in the temporary or permanent impairment of motor, sensory, and autonomic functions [1]. These impairments significantly affect the impacted individual’s functional ability, psychological well-being, and ability to engage in daily activities and social interactions [2,3,4]. Community participation (CP) following SCI/D can be challenging due to the associated primary impairments and secondary health conditions, as well as difficulties navigating both built and social–emotional environments [5]. CP is a multifaceted concept that is influenced by an individual’s perceptions and relates to their social health and well-being, engagement, a sense of agency at both the personal and societal levels, and social connections [6]. Routine clinical assessment of CP (self-efficacy and return to normal life) is crucial for determining the effectiveness and success of rehabilitation. [7,8]. Most often, ‘participation’ is defined by the International Classification of Functioning, Disability and Health (ICF) as ‘involvement in a life situation’ [9].

CP and social reintegration after SCI/D are drastically affected by the built environment in urban and rural areas [10]. The CDC has reported that “rural residents face higher risks of death due to limited access to specialized medical care and emergency services and exposure to specific environmental hazards” [11], such as forest fires or episodic loss of power. Consequently, rural residents may migrate to urban areas after injury to increase access to attendant care or specialty medical services [12]. Individuals living in socioeconomically disadvantaged environments such as rural or low-income areas tend to have different rehabilitation outcomes than those who do not face such challenges [13]. Further, prior research has highlighted significant differences in barriers to care between urban and rural areas due to differences in access to healthcare, rehabilitation, and support services, potentially affecting the individual’s health and well-being [14,15]. Thus, the geographic setting in which a person resides has been widely considered a key determinant of CP in the SCI/D population.

The severity of an injury depends significantly on the extent of spinal cord damage and completeness of the injury [16]. The ASIA Impairment Scale (AIS) is a fundamental tool in the assessment and classification of SCI severity and offers a standardized method for reporting the extent and completeness of SCI [17,18], which facilitates the consistent evaluation and comparison of outcomes across studies and therapeutic interventions [19]. Individuals with motor complete SCI (AIS A–B) typically show limited and predictable recovery, whereas those with incomplete SCI (AIS C and AIS D) demonstrate a more substantial and variable recovery potential [20]. Therefore, the severity of SCI/D is of paramount importance in determining CP among individuals living with SCI/D.

In individuals with SCI/D, the Moorong Self-Efficacy Scale (MSES) and Reintegration to Normal Living Index (RNLI) are valuable tools for evaluating and planning rehabilitation processes [21] and assessing CP and reintegration into society [22]. The MSES is a validated tool for individuals with SCI/D to measure their perceived ability to perform various activities such as self-care, leisure activities, and emotional well-being [21,23,24]. The RNLI is a tool that uses interviews or self-reported survey responses to evaluate one’s satisfaction with their ability to return to pre-injury daily activities and participate in their community [25]. The RNLI tool assesses mobility, self-care, leisure pursuits, engagement in community activities, and interpersonal connections with others [5]. The collection of CP indicators of quality care is easy, yet time-consuming, as the process requires self-reported surveys prior to inpatient rehabilitation discharge and the collection of data in transitional living and community settings.

In this context, geographical location (e.g., urban vs. rural) has been widely considered a likely determinant of health-related quality of life in SCI/D populations. However, investigations on this topic remain limited, particularly with respect to the AIS categories.

The implementation of technology and computerization of healthcare surveys may enhance the speed and accuracy of data collection, extraction, and reporting [26], particularly in the evaluation of CP. To assess CP, AI-based optical mark recognition (OMR) software may help reduce the time required for manual survey data extraction [27]. Such a strategy could also foster more informed and timely decision making based on the information obtained from multiple contrasting community settings.

The investigation into urban–rural disparities in CP among individuals with lived experience with SCI/D in Ontario addresses a critical gap in our understanding of how geographical and environmental factors influence the lives of those with SCI/D. This question is particularly pertinent given the unique challenges faced by individuals with lived experience navigating both physical and social environments. By examining the differences between urban and rural residents, this quality improvement project aims to uncover how varying levels of accessibility, available resources, and social support systems influence CP. This exploration is crucial for identifying specific barriers and facilitators to CP in each environment, to inform targeted interventions and policy recommendations. Quality outcomes from this project may have broader implications for healthcare planning and social service provision across diverse geographical areas and, ultimately, address the larger goal of our network to promote optimal and equitable care for all Canadians with SCI/D, regardless of their residential location.

Therefore, this quality improvement project seeks to describe and compare self-reported CP (RNLI and MSES) scores of individuals with motor complete and incomplete SCI/D living in urban or rural Ontario, Canada; and 2) to determine the accuracy of an artificial AI OMR tool for extracting data from CP surveys conducted among participants transitioning from inpatient rehabilitation to home and residing in the community.

## 2. Methodology

### 2.1. Study Population

Between January and October 2022, survey data were collected using an iPad Pro (Apple, CA, USA) and UnStackr (OMR software version 6.7.5 available at https://www.unstackr.com/ (accessed on 10 September 2024), formerly known as Reachlite^TM^) from adults with SCI/D living in Ontario, Canada. Convenience sampling was used to collect demographic, RNLI, and MSES survey data from adults at least 18 months post-SCI/D onset with a neurological level of injury from C1-L5 AIS A-D. Consenting individuals were current or former inpatients affiliated with the University Health Network, Lyndhurst Centre (LC) (i.e., Canada’s largest free-standing SCI rehabilitation facility, a tertiary academic rehabilitation hospital within the University Health Network, and a member of the Toronto Academic Health Science Network [28]) or Spinal Cord Injury Ontario (SCIO). SCIO is a community service organization that seeks to ensure that individuals with spinal cord injury “*live the life they choose in a fully inclusive Ontario*” through the provision of service navigation and peer mentor services [29].

### 2.2. Survey and Tools

The survey (Appendix A) was designed specifically for the assessment of the ‘Community Participation’ domain [5] of the Spinal Cord Injury Implementation and Evaluation Quality Care Consortium (SCI-IEQCC) [30]. The survey comprised demographic and impairment data and the MSES and RNLI surveys.

The MSES is a self-report questionnaire that assesses two areas of functioning: daily activities (e.g., personal hygiene) and social functioning (e.g., enjoying time with friends) [23]. Participants are asked to rate their confidence in completing 16 tasks/items on a seven-point Likert scale, ranging from “0”—“very uncertain” to “7”—“very certain.” MSES scores range from 16 to 112, with a higher score indicating greater self-efficacy or confidence in one’s ability to control one’s behavior and outcomes [21]. The time for MSES completion was five minutes.

The RNLI includes 11 items, each with a visual analog scale (VAS) that allows individuals to gauge the extent to which the statement applies to their specific situation [25]. The VAS is anchored by phrases that describe the patient’s situation and is scored on a scale of 0–10, with 0 representing minimal reintegration and 10 representing complete reintegration. The index ranges between 0 and 110, and the summary score is then adjusted to a maximum of 100 ($\frac{Total\;RNLI\;score}{110}\times 100$). Alternate variations of RNLI are 3- and 4-point scoring systems, which have been validated for data collection over the phone [31,32]. The time for RNLI completion was approximately 10 minutes.

### 2.3. Data Collection Methods

Adults with SCI/D living in the community completed the survey (see Appendix A). Survey responses were recorded on paper or in digital format using an iPad Pro and Apple Pencil^®^. At UHN, a Research Associate collected the data via in-person interview or telephone interview. For outpatients affiliated with SCIO, a provincial intake coordinator collected data via phone interview. The intake coordinator made a maximum of three attempts to contact potential participants by phone to obtain verbal consent for participation. The procedures for data collection and processes for responding to patient survey responses were standardized across participating organizations. Verbal consent was obtained from all participants prior to data collection. All data were de-identified, scanned, and then pooled at University Health Network for statistical analyses. Research ethics waivers and/or QI approvals were obtained for the project (QI ID#: 20-0007), and appropriate data-sharing agreements and confidentiality agreements were established.

### 2.4. AI-Based OMR Tool

Following data collection, the digitized versions of completed surveys (i.e., scanned paper-based surveys using the 300-dpi setting of a professional office scanner by KONICA MINOLTA [bizhub 458e] or the portable document format [pdf] of iPad-based surveys) were imported into OMR software, which was custom designed for the intended purpose of collecting data across the continuum of care. UnStackr allows the operator to extract survey responses from a scanned image using its algorithms. As the responses are processed, UnStackr extracts data, recognizes the selected response, and associates the data with predefined data elements in the structured JSON format. The AI tool then maps the data from JSON format and pushes the data into a comma-separated value output file (i.e., CSV file that can be opened by Microsoft Excel^®^) for local storage, validation for accuracy of extraction, and statistical analysis.

### 2.5. Rurality Index of Ontario

The Rurality Index of Ontario (RIO) score was developed by the Ontario Ministry of Health and Long-Term Care and the Ontario Medical Association [33,34]; to calculate RIO scores, community population and density, travel time to the nearest basic referral center, and travel time to the nearest advanced referral center were considered [34]. The score was calculated for all census subdivisions of Ontario and ranged from 0 to 100, with three stratifications: major urban (0–9), non-major urban (10–40), and rural (>40) areas [34]. These classifications are more intuitive for policymaking, research purposes, and healthcare assessments and are routinely used in health service research in Ontario to describe and to compare urban versus rural health system resources. Additionally, the RIO is increasingly being used as an eligibility factor for incentive programs offered to rural physicians to address gaps in service.

### 2.6. Statistical Analysis

Descriptive statistics were used to summarize the demographic and impairment characteristics and mobility devices of the MSES and RNLI respondents, data collection tools, and the accuracy of the extracted data. We used non-parametric tests (e.g., Mann–Whitney) to compare MSES and RNLI in individuals with AIS categories A–B versus C–D, stratified by their RIO or residential status (i.e., major urban vs. rural or non-major urban areas). Rank biserial correlations were used to evaluate the associations between MSES and RNLI with AIS categories in each residential stratum [35].

## 3. Results

Eighty-five individuals (mean age 53.7 years, standard deviation, 14) with SCI/D responded to our CP survey (76.5% UHN-LC). The majority were men (63.5%), had a duration of SCI/D of more than 10 years (44.7%), were from major urban areas (69.4%), and had incomplete tetraplegia (AIS C–D, 40.0%). The most common mobility devices in the community and at home were power (36.5%) and manual (37.6%) wheelchairs, respectively. The CP surveys were collected digitally from approximately 75% of our respondents (see Table 1).

### 3.1. MSES and RNLI

The mean total MSES score for Ontarians with SCI/D was 87.96 (95% confidence interval [CI]: 84.45, 91.47). In the MSES, item 6, ‘I can have a satisfying sexual relationship’ (mean: 4.012, 95% CI: 3.527–4.497), had the lowest scores, while item 4, ‘I can maintain relationships in my family’ (mean: 6.494, 95% CI: 6.275–6.713), had the highest scores among respondents (Appendix A).

For the same individuals, the mean total RNLI score was 75.61 (95% CI: 71.85, 79.37). Item 3, ‘I am able to take trips out of town as I feel are necessary’ (5.965, 95% CI: 5.252–6.678), had the lowest RNLI domain score, whereas item 1, ‘I move around my living quarters as I feel is necessary’ (8.447, 95% CI: 8.008–8.887), had the highest domain score (Table 2).

### 3.2. MSES and RNLI by Impairment Levels

The MSES for participants in the AIS A–B category (*n* = 25) had a mean score of 95.92, whereas the MSES for those in the AIS C–D category (*n* = 60) had a mean of 84.65 (Figure 1a). The results of the non-parametric test ($W_{Mann-Whitney}=1050,\;\;p=3.80\times 10^{-3}$) showed that individuals in AIS categories C–D generally had significantly lower MSES scores than those in the lower AIS A–B impairment category. On the other hand, the mean RNLI score for the AIS A–B category (*n* = 25) was 80.36, while the mean RNLI score for the AIS C–D category (*n* = 60) was 73.63. Although these results implied that individuals with incomplete impairment exhibited lower RNLI scores than those in the AIS A–B category, this difference was not statistically significant ($W_{Mann-Whitney}=925,\;p=0.09$) (Figure 1b).

### 3.3. MSES and RNLI in Major Urban Areas

The mean MSES score for the AIS A–B category (*n* = 19) was 92.79, while the mean MSES for the AIS C–D category (*n* = 40) was 85.83 (Figure 2a), which were not significantly different according to AIS categories A–B and AIS C–D among participants living in major urban areas ($W_{Mann-Whitney}=459,\;p=0.2$). The rank-biserial correlation (0.21, 95% CI:−0.11, 0.48; n=59) identified a small relationship between MSES score and AIS category in individuals living with SCI/D and residing in major urban areas.

The mean RNLI scores for the AIS A–B and AIS C–D categories were 77.74 and 76.40 (Figure 2b), respectively, and were not significantly different ($W_{Mann-Whitney}=400,\;p=0.75$). The rank-biserial correlation (0.05, 95% CI:−0.26, 0.35; n=59) suggested no significant unidirectional association between RNLI score and AIS category in participants with SCI/D residing in major urban areas.

### 3.4. MSES and RNLI in Rural or Non-Major Urban Areas

In rural or non-major urban areas, the analysis indicated a statistically significant difference in MSES with an average for the AIS A–B category (*n* = 6) equal to 105.83, and for the AIS C–D category (*n* = 20) equal to 82.30 ($W_{Mann-Whitney}=111,\;p=0.00211$; Figure 2c). The average RNLI scores in the AIS A–B and AIS C–D categories were 88.67 and 68.10 respectively, which were significantly different between the two AIS category categories ($W_{Mann-Whitney}=101.5,\;p=0.01$; Figure 2d).

The rank-biserial correlation between the MSES (0.85, 95% CI: 0.62, 0.95; n=26) or RNLI score (0.69, 95% CI:0.30, 0.88; n=26) and AIS categories confirmed the strong associations between MSES or RNLI scores and AIS categories in rural or non-major urban areas.

### 3.5. Results from AI Solution for Data Extraction

Validation of the extracted data from the AI system showed promising results because the error rate was consistently below 5%. The threshold set at the beginning of the pilot testing was 2.59% for errors in the total number of data fields (88/3400), whereas the error rate was 0.26% when the total data fields per form were considered (see Table 3).

## 4. Discussion

This study evaluated CP using the RNLI and MSES among individuals living with complete or incomplete SCI/D who resided in major urban, non-major urban, or rural regions in Ontario, Canada. The study results revealed evident distinctions in the RNLI and MSES scores among individuals with AIS C and D compared to AIS A and B who live in rural or non-major urban settings, whereas such disparities were not observed among those residing in major urban areas.

The MSES serves as a valuable tool for assessing self-efficacy trajectories and guiding intervention strategies to enhance individuals’ confidence in their abilities [36,37]. Middleton, Tran [21] included persons with SCI/D from Australia (mean age: 48.6, SD = 16.8 years) and the United States (mean age: 48.5, SD = 13.1 years), with the majority being male (79% among Australians and 67% among Americans) and paraplegic (54% of Australian responders and 53% of Americans). The lowest domain of self-efficacy in this study was observed in item 6, ‘I can have a satisfying sexual relation’, (3.18 ± 2.32), which is consistent with the findings of our study (4.01, 95% CI: 3.53, 4.50). Sexual dysfunction is a major concern in both men and women with SCI; however, it is often overlooked in rehabilitation programs [38]. Sexual health is a common unmet need in the rehabilitation of individuals with SCI/D, and sexual function is among the top priorities for individuals with SCI/D, regardless of the level of injury or time since injury [39]. Additionally, according to findings by Alavinia and colleagues [40], sexual health was a top priority for Canadians living with SCI/D, which was based on the perceived high importance and low feasibility of this domain aim. However, despite the high prevalence of sexual dysfunction in individuals with SCI/D, many rehabilitation programs lack comprehensive sexual health education and treatment programs [41].

Similarly, the highest MSES score reported by Middleton, Tran [21] was in item 4, ‘I can maintain relations in my family’ (6.08 ± 1.41), which aligned with our findings (6.49, 95% CI: 6.28, 6.71). The overall MSES score in a prior Australian cohort of individuals with SCI/D [7] was 84.5 (95% CI: 80.4, 88.6), which was slightly lower than the total scores reported in our study (87.96, 95% CI: 84.45, 91.47). The MSES scores are consistent with prior evidence, although the differences in age, duration of injury, and proportion of paraplegics among the Canadian and Australian cohort members may explain the observed differences [7].

An Italian study which recruited 65 adults with SCI (41 men, 24 women), the majority of whom were paraplegic (71.9%) with motor complete injuries (60.7%), with a mean age of 55.4 (SD = 14.3 years) and mean time since injury of 26 years (SD = 20.3 years), reported the average MSES score to be 98.51 (SD = 12.41), which was significantly higher than our findings (87.96, 95% CI: 84.45, 91.47). However, the study included a higher proportion of paraplegics (71.9% vs. 50.6%), and the mean time since injury was significantly higher than that in the current study (26 years, SD = 20.3). These differences affirm the higher MSES scores reported among individuals living with motor complete paraplegia who are wheelchair dependent.

To date, validated CP measures, such as RNLI, years post-injury, impairment, ambulatory status, employment, and health status, are all significant predictors of community participation [31]. Factors influencing CP include employment, living situation, sex, and the presence of muscle spasms [42,43], which can restrict motor functionality in individuals with SCI/D. A previous study in Canada validated a 3-point scale version of the RNLI tool [31]. The authors reported that item 6, ‘I am able to participate in recreational activities as I want to’, was the most challenging (165/618 participants selected ‘Does not describe my situation’), whereas in our study, item 3, ‘I am able to take trips out of town as I feel are necessary’, had the lowest score (mean 5.97, 95% CI: 5.25, 6.68). In both studies, item 1, ‘I move around my living quarters as I feel necessary’, had the highest scores.

These responses suggest that rehabilitation processes prepare patients well for household mobility but less so for leisure and recreational activities, including travel. These specific service gaps are worthy of substantive best-practice implementation efforts. Therefore, travel-related concerns in rehabilitation to improve the quality of life and participation in society for individuals with SCI/D need to be underscored [44,45] and addressed in future health system advancements.

In general, individuals with higher physical independence and greater functional abilities typically experience fewer perceived barriers to CP [46]. The enclosed survey results suggest that an individual’s mobility may correlate with their perception of environmental barriers and that the barriers perceived by marginal community ambulators versus community wheelchair users are different. Furthermore, environmental barriers differ in rural and non-major urban areas, posing unique challenges for individuals with AIS C–D impairment who would typically walk short distances with gait aids outside the home. The importance of environmental accessibility and resources in facilitating social participation [44,45] suggests that the typically less accessible infrastructure in rural areas may act as a barrier to community engagement for individuals with SCI/D, disproportionally impacting those with AIS C–D impairment. Further, the perception of self-efficacy may be different for those with a visible motor complete disability, as opposed to those with a motor incomplete and less visible or invisible disability.

Noreau, Noonan [47] conducted a comprehensive Canadian national survey, known as the Spinal Cord Injury Community Survey. The study aimed to collect data on various aspects of life for people with SCI, including data describing their health, well-being, CP, and employment. The survey was designed to address gaps in knowledge about the SCI population in Canada and to inform policy and service development. The study found that CP levels differed between individuals with SCI living in urban and rural areas. Those living in urban areas generally reported higher levels of CP compared to their rural counterparts. This difference was attributed to several factors, including better accessibility in urban environments, more diverse transportation options, and a greater variety of community programs and services available in cities.

Rural residents with SCI faced additional challenges in accessing community activities and services due to geographical barriers, limited transportation options, and fewer adapted facilities. Additionally, the authors noted that rural communities often provided stronger social support networks, which could partially compensate for the reduced formal services available. However, rural areas pose unique challenges, such as limited access to accessible transportation, sidewalks, and wheelchair trails; fewer global regional resources; and the reduced availability of specialized services, potentially hindering CP.

Thus, the presence of AIS C–D impairment but presumed ambulation does not guarantee CP in such environments. For example, despite having motor function, individuals with SCI/D may still face participation restrictions owing to environmental barriers, as observed in the Swiss community [46]. Urban residents with SCI/D tend to have greater access to specialized care, resulting in higher healthcare utilization rates, whereas rural residents may face barriers, such as the distance to healthcare facilities and the limited availability of specialized services [48,49]. Therefore, limited access to healthcare facilities may result in the decision to relocate from rural to urban areas post-injury, underscoring the importance of access to healthcare infrastructure in shaping individuals’ choices of residence [50]. These disparities echo the findings of a systematic review by Kashif, Jones [51], who identified health-related, personal, environmental, psychological, and social barriers to community reintegration in individuals with SCI/D. This review highlights the importance of removing these barriers to enhance CP [51].

The AI tool for data extraction was validated against trained research staff, which revealed that the error rate was acceptable, considering the time needed for data extraction (<10 s per full survey questionnaire). This aligns with findings from other studies that have explored the use of AI in research data processing. For instance, Joseph, Lindblad [52] tested a Customized eXtraction Program (CXP, version 4.69, IQVIA Stockholm, Sweden), which was software for the extraction of data from electronic medical records. The authors reported that their custom-developed software had accuracy of 97.5% in correctly identifying the requested data. The use of the OMR tool is promising for facilitating rapid and accurate data extraction from surveys, and the tool is ready for community deployment on a national scale. Our reported accuracy of the OMR tool aligns with a previous study that found the accuracy rate of the extracted data was more than 99% [53]. This speed and accuracy may help drive the creation of tailored reports that not only provide on-demand analysis of the data, but also include guidelines to address the needs of the individual in the areas of self-efficacy and reintegration into the community among individuals living with SCI/D. This would ultimately help to identify gaps in rehabilitation and push local providers to create local CP solutions for those with identified low CP survey scores.

The neurological level of injury and AIS determines the extent of motor function impairment and affects disability and CP, with a notable decline in participation among individuals with cervical injuries [54] versus those with thoracic or lumbar neurologic levels. This highlights the importance of motor function and mobility, which are crucial components of an individual’s functional abilities and play pivotal roles in CP in individuals with SCI/D [55]. Overall, the QI project findings highlight the significant role of rehabilitation processes in shaping the CP outcomes of individuals with SCI/D who have AIS A–B versus AIS C–D impairments.

## 5. Limitations

The data collection process may have biased the survey results, with respondents able to self-complete the iPad surveys, whereas telephone surveys were conducted by SCIO trained staff, which may have increased the respondents’ reluctance to answer honestly. The power of our analysis is likely to be improved by using a larger sample size to obtain more statistically robust results. Additionally, the data collection methods employed at the UHN and SCIO differed considerably. Finally, data collection and subsequent analysis did not account for potential seasonal variations in the RNLI and MSES scores. However, it is plausible that individuals’ perceptions and responses may have been influenced by seasonal and weather-related environmental factors such as the presence of snow, rainfall, or extreme heat. These environmental conditions could affect an individual’s mood, cognitive processes, and overall well-being, thereby potentially affecting their self-reported RNLI and MSES scores.

### Opportunities for Improvement

The World Health Organization’s guidelines on community-based rehabilitation (CBR) emphasize the importance of addressing the full spectrum of needs of individuals with disabilities, including often overlooked areas such as sexual health and community participation [56]. Community-based programs can effectively address the many unmet needs of individuals with SCI/D, including those related to sexuality and social participation. There is a compelling need to enhance CP through the implementation of community-based rehabilitation concepts and a deep understanding of the individual’s lived experience.

## 6. Conclusions

MSES and RNLI can characterize CP and identify service gaps among individuals with SCI/D. There is an opportunity to enhance SCI rehabilitation programs to ensure sexual satisfaction and the ability of individuals with SCI/D to participate in recreational and leisure pursuits and travel. There are unique unmet needs for marginalized individuals residing in rural or non-major urban areas with incomplete SCI/D (AIS category C or D) to enhance CP. Furthermore, the OMR tool for extracting data from survey questionnaires demonstrated a reliable level of accuracy and enhanced efficiency in terms of speed for future use across urban and rural settings to monitor CP over time among individuals with SCI/D.

## Figures and Tables

**Figure 1 healthcare-12-02089-f001:**
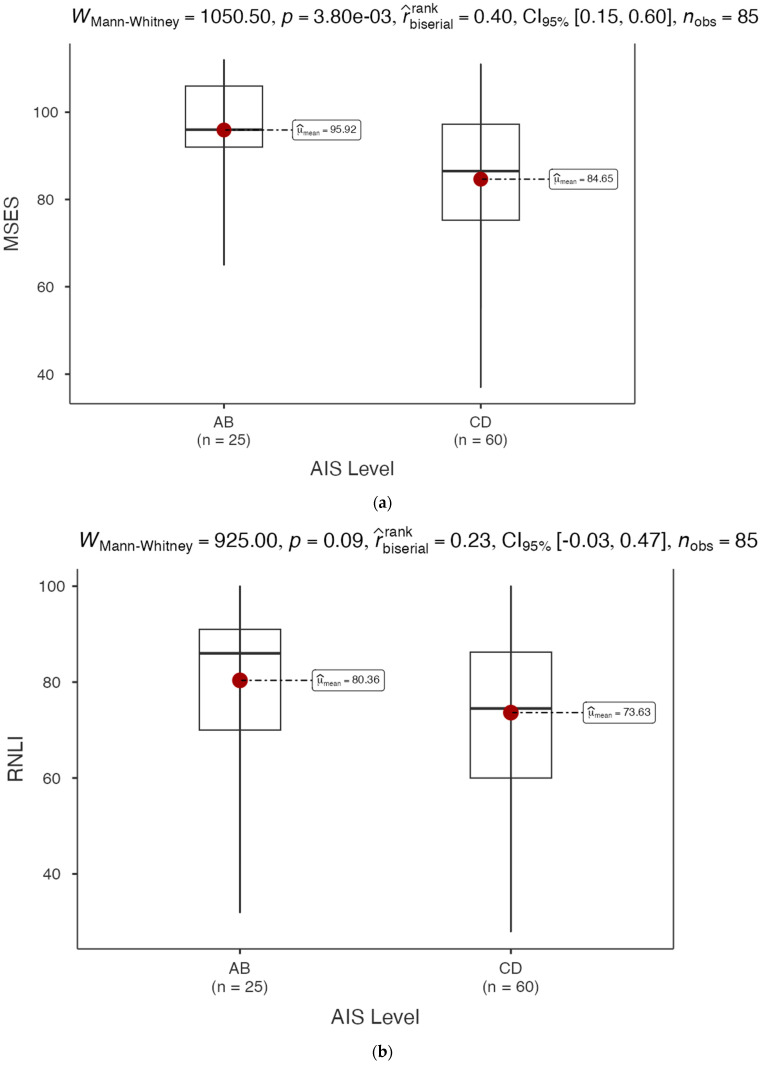
(**a**) MSES total score by impairment level for CP survey respondents; (**b**) RNLI total score by impairment level for CP survey respondents.

**Figure 2 healthcare-12-02089-f002:**
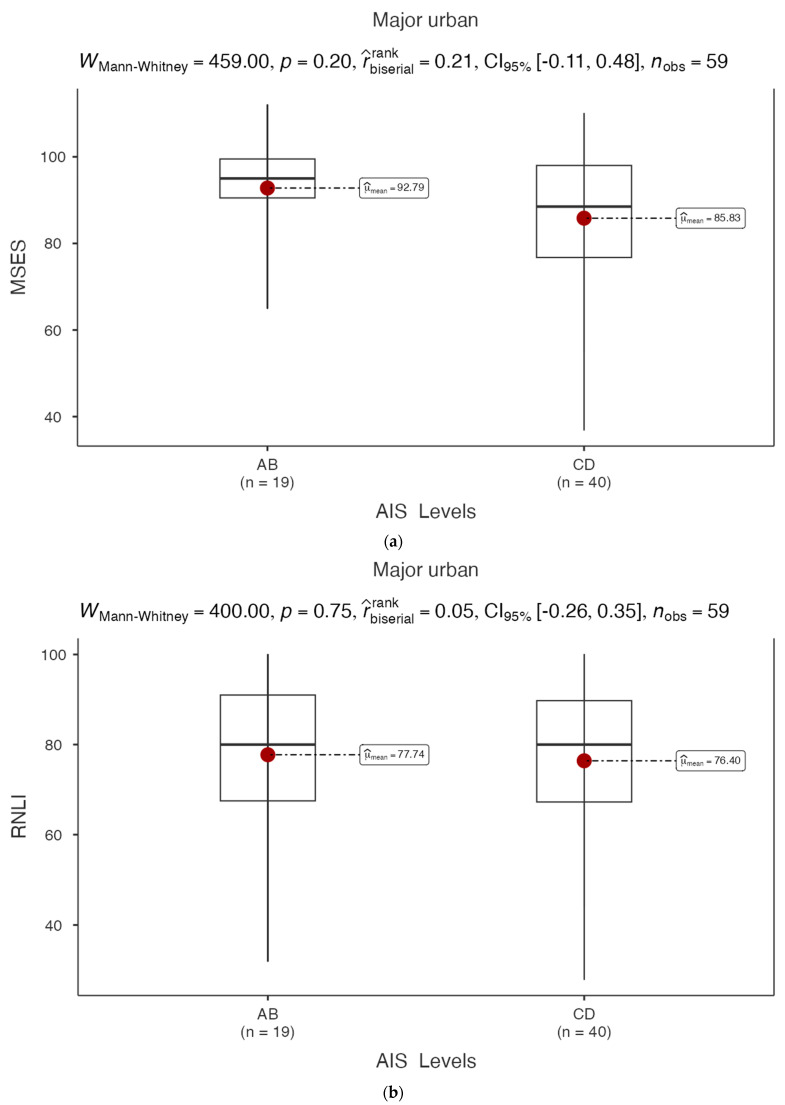
(**a**) MSES total score by impairment level in major urban CP survey respondents; (**b**) RNLI total score by impairment level in major urban CP survey respondents; (**c**) MSES total score by impairment level in rural or non-major urban CP survey respondents; (**d**) RNLI total score by impairment level in rural or non-major urban CP survey respondents.

**Table 1 healthcare-12-02089-t001:** Demographic and impairment characteristics of CP survey respondents (N = 85).

Characteristic	
**Age, mean (SD)**	53.7 (14.0)
**Sex, *n* (%)**	
Female	31 (36.5%)
Male	54 (63.5%)
**Duration of Injury, *n* (%)**	
<2 years	14 (16.5%)
2–10 years	33 (38.8%)
>10 years	38 (44.7%)
**Impairment, AIS Category, *n* (%)**	
Paraplegia A–B	17 (20.0%)
Paraplegia C–D	26 (30.6%)
Tetraplegia A–B	8 (9.4%)
Tetraplegia C–D	34 (40.0%)
**Mobility Devices—Home, *n* (%)**	
Cane	3 (3.5%)
Manual wheelchair	32 (37.6%)
Power wheelchair	27 (31.8%)
Walker	8 (9.4%)
Walking without assistance	13 (15.3%)
Other	2 (2.4%)
**Mobility Devices—Community, *n* (%)**	
Cane	6 (7.1%)
Manual wheelchair	27 (31.8%)
Power wheelchair	31 (36.5%)
Walker	11 (12.9%)
Walking without assistance	4 (4.7%)
Other	6 (7.1%)
**Rurality, *n* (%)**	
Major urban	59 (69.4%)
Rural or non-major urban	26 (30.6%)
**Source of Data Collection, *n* (%)**	
Lyndhurst **Centre**	65 (76.5%)
Spinal Cord Injury Ontario	20 (23.5%)
**Data Collection Method, *n* (%)**	
Paper	21 (24.7%)
iPad Pro^®^	64 (75.3%)

AIS, ASIA Impairment Scale; SD, standard deviation.

**Table 2 healthcare-12-02089-t002:** MSES and RNLI item responses across all survey respondents (N = 85).

	Items	Mean (SD)	95% CI
**MSES**	Personal hygiene	5.85 (1.78)	5.47, 6.23
	Bowel accidents	5.54 (1.68)	5.18, 5.9
	Active household member	5.53 (1.68)	5.17, 5.89
	Relationships in family	6.49 (1.03)	6.28, 6.71
	Get out of house	5.53 (1.91)	5.12, 5.94
	Sexual relationship	4.01 (2.28)	3.53, 4.5
	Time with friends	5.58 (1.76)	5.2, 5.95
	Hobbies and leisure pursuits	5.41 (1.82)	5.03, 5.8
	Contact with important people	6.26 (1.12)	6.02, 6.5
	Deal with unexpected problems	5.64 (1.31)	5.36, 5.91
	Work in future	4.15 (2.43)	3.64, 4.67
	Can accomplish most things	5.42 (1.61)	5.08, 5.77
	Learn something new	5.88 (1.23)	5.62, 6.14
	Make first contact	5.35 (1.75)	4.98, 5.73
	Maintain good health	5.78 (1.33)	5.49, 6.06
	Fulfilling lifestyle in future	5.54 (1.62)	5.2, 5.89
	MSES Total Scores	87.96 (16.51)	84.45, 91.47
**RNLI**	Move around living quarters	8.45 (2.07)	8.01, 8.89
	Move around community	7.04 (2.5)	6.50, 7.57
	Able to take trips	5.96 (3.35)	5.25, 6.68
	Comfortable with self-care	7.92 (2.53)	7.38, 8.46
	Spend most days at work	6.29 (3.19)	5.62, 6.97
	Participate in recreational activities	7.74 (2.23)	7.27, 8.22
	Participate in social activities	7.42 (2.41)	6.91, 7.94
	Assume role in family	7.69 (2.29)	7.21, 8.18
	Comfortable with personal relationships	8.41 (1.75)	8.04, 8.79
	Comfortable in company of others	8.22 (1.93)	7.81, 8.63
	Deal with life events	8.04 (1.78)	7.66, 8.41
	RNLI Total Scores	75.61 (17.68)	71.85, 79.37

CI, confidence interval; MSES, Moorong Self-Efficacy Scale; RNLI, Reintegration to Normal Living Index.

**Table 3 healthcare-12-02089-t003:** Data extraction errors for the community participation domain.

Item	Value
Number of Questions	
Number of forms	85
Number of questions	40
Total data fields (# forms × # questions)	3400
Total errors	88
Percentage errors	2.59%
Individualized Items	
Number of data fields per form	403
Total data fields (# forms × # data fields/form)	34,255
Total errors	88
Percentage errors	0.26%

## Data Availability

The data presented in this study are available upon request from the project leader and corresponding author Dr. Beverley Catharine Craven.

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
