# Peer review of "Urban–Rural Disparities in Community Participation after Spinal Cord Injury in Ontario"

_healthcare, 2024, doi:10.3390/healthcare12202089_

Round 1

Reviewer 1 Report

Comments and Suggestions for Authors

This paper is really interesting and valuable. The challenge of a full social recovery for SCI patients is ever current all over the world.

Some concerns have to be better addressed and corrected.

From a methodological point of view, the survey presents many possible biases itself.  The way of distribution should be better described. How many people did the initial sample consist of and how many took part in the survey at the end? Did you carry out a sample size calculation? An istitutional review board could be useful, did you obtain it? Why did you collect the informed consent just in a verbal way? Why data were collected in two different ways? A clear definition of rural and urban provenience is missing. Please clarify all these aspects. 

As regards the social inclusion for people suffering from a SCI, a great importance lies in sporting activities. This aspect should be deepened in the discussion, and to do that I suggest the following reference:

-Farì, G., Ranieri, M., Marvulli, R., Dell'Anna, L., Fai, A., Tognolo, L., Bernetti, A., Caforio, L., Megna, M., & Losavio, E. (2023). Is There a New Road to Spinal Cord Injury Rehabilitation? A Case Report about the Effects of Driving a Go-Kart on Muscle Spasticity. Diseases (Basel, Switzerland)11(3), 107. https://doi.org/10.3390/diseases11030107

Best regards

Author Response

Response to Reviewer 1: We sincerely appreciate the insightful feedback from Reviewer 1 and the opportunity to address the concerns. We have implemented the following comprehensive modifications to the manuscript:

Data collection:

Thank you for your invaluable insights! We provided further details about the data collection processes in sections 2.3 and 2.4.

Quality Improvement (QI) project, institutional review board approval, and participation consent:

It is important to note that this QI initiative was undertaken by health system leaders, clinicians, and patients affiliated with University Health Network (UHN) and SCI Ontario. Research ethics waivers and/or QI approvals were obtained for the project, and appropriate data-sharing agreements and confidentiality agreements were established between participating organizations (UHN QI # 20-007). Participants provided verbal consent for participation. Noteworthy is that QI projects are a means of:

  • Building a common understanding of service gaps in a specific community or health care sector
  • Promoting the spread of best practices or promising health service enhancements across agencies and providers 
  • Building capacity to use data within a health sector to inform the standard of care and support comparisons across organizations
  • Driving organizational readiness for change, including performance measurement and reporting

Definition of rural and urban provenience:

Thank you for your suggestion! In the manuscript (see section 2.5), we elaborate that the Rurality Index for Ontario (RIO) score is a measure of rurality that ensures funding is specifically targeted to northern and very rural communities. The RIO2008_Basic score is derived from three factors:

  • population (count and density)
  • travel time to a basic referral centre
  • travel time to an advanced referral centre

RIO scores are assigned to Statistics Canada census subdivisions (CSDs). The RIO scores are routinely used in health service research in Ontario to describe and compare urban versus rural health system resources.

Community Participation versus physical/sporting activities after SCI/D:

We appreciate providing the reference to improve our discussion (Farì, G., Ranieri, M., Marvulli, R., Dell'Anna, L., Fai, A., Tognolo, L., Bernetti, A., Caforio, L., Megna, M., & Losavio, E. (2023). Is There a New Road to Spinal Cord Injury Rehabilitation? A Case Report about the Effects of Driving a Go-Kart on Muscle Spasticity. Diseases (Basel, Switzerland), 11(3), 107. https://doi.org/10.3390/diseases11030107). However, though the article is interesting upon review it was deemed not relevant to the context of our manuscript since the suggested article demonstrated promising effects of adaptive sports, such as go-kart driving, on muscle spasticity in SCI patients, whereas we investigated the community participation after SCI/D and its disparities between urban and rural areas. Nonetheless, we expanded the Discussion section and included previously published literature such as Noreau et al (2014).

We thank you again for your invaluable feedback on our paper.

Reviewer 2 Report

Comments and Suggestions for Authors

This is a comprehensive manuscript and it is worth for publication in this journal. However, minor modifications need to be done before can be considered for acceptance.

1. Introduction

a) Lines 80-95: I cannot understand the relationship between a sentence of AI with the current research finding. If the authors are doing AI method, then it is relevant. However, the authors only use AI-tool to collect the data.

b) Unclear research gap: It should be noted that authors should highlight previous drawbacks of the findings that led to this current study. A research gap should be clearly highlighted before authors can come out with the strong objectives of the research.

2. Methodology

a) Verbal consent is not strong enough to support this research as normally a researcher should provide written consent. I understand it is difficult to collect all written forms if the participation is many, but if possible, please include some recorded audio of the participants who agreed to participate, as an appendix.

b) Have the authors tried many AI-tools to extract data? Please include the pilot findings before decide the best AI-tools. Otherwise, the results are not promising.

c) A scanner is another issue of inconsistency in data extraction. Please detail out the brand and model. If possible, some technical data is needed to justify the use of the current scanner in the manuscript.

d) Participant recruitment is not clear. How 85 participants were selected?

3. Results

I have no issue with the results since it looks promising and giving new insights.

4. Discussion

a) It is better to include a discussion about the use of AI-tool and scanner in terms of accuracy, time taken, and consistency of the data collection.

Author Response

Response to Reviewer 2: We are grateful for your feedback on our manuscript. We have implemented the following comprehensive modifications to the manuscript:

  1. Introduction
  2. a) Lines 80-95: I cannot understand the relationship between a sentence of AI with the current research finding. If the authors are doing AI method, then it is relevant. However, the authors only use AI-tool to collect the data.

Thank you for your comment! We tailored and moved the revised paragraph to clarify the use of the AI-based tool for data extraction from the collected surveys.

  1. b) Unclear research gap: It should be noted that authors should highlight previous drawbacks of the findings that led to this current study. A research gap should be clearly highlighted before authors can come out with the strong objectives of the research.

Thank you for the comment! We added a clearer research gap in the Introduction.

  1. Methodology
  2. a) Verbal consent is not strong enough to support this research as normally a researcher should provide written consent. I understand it is difficult to collect all written forms if the participation is many, but if possible, please include some recorded audio of the participants who agreed to participate, as an appendix.

We appreciate this comment. Please let us elaborate further that this project was a Quality Improvement (QI) study. It is important to note that this QI initiative was undertaken by health system leaders, clinicians, and patients affiliated with University Health Network (UHN) and SCI Ontario. Research ethics waivers and/or QI approvals were obtained for the project, and appropriate data-sharing agreements and confidentiality agreements were established between participating organizations (UHN QI # 20-007). Participants provided verbal consent for participation. Noteworthy is that QI projects are a means of:

  • Building a common understanding of service gaps in a specific community or health care sector
  • Promoting the spread of best practices or promising health service enhancements across agencies and providers 
  • Building capacity to use data within a health sector to inform the standard of care and support comparisons across organizations
  • Driving organizational readiness for change, including performance measurement and reporting

  1. b) Have the authors tried many AI-tools to extract data? Please include the pilot findings before decide the best AI-tools. Otherwise, the results are not promising.

We used and evaluated Reachlite, which was a custom-made AI-based optical mark recognition (OMR), for data extraction from CP surveys. We did not pilot tested any other AI tools for data extraction.

  1. c) A scanner is another issue of inconsistency in data extraction. Please detail out the brand and model. If possible, some technical data is needed to justify the use of the current scanner in the manuscript.

We included further details about the scanner and setting used in section 2.4. 

  1. d) Participant recruitment is not clear. How 85 participants were selected?

Thank you for your question! This study was a QI project as elaborated in question 2a and therefore rigorous research specific processes such as pre-determined sample size calculations were not applicable to this QI study.

  1. Results

I have no issue with the results since it looks promising and giving new insights.

Thank you for this comment!

  1. Discussion
  2. a) It is better to include a discussion about the use of AI-tool and scanner in terms of accuracy, time taken, and consistency of the data collection.

Thank you for your comment! We provided further discussion pertaining to the AI-tool in the Discussion section.

We thank you again for your invaluable feedback on our paper.

Round 2

Reviewer 1 Report

Comments and Suggestions for Authors

Thank you for the efforts to improve the quality of your paper according to my suggestions.

It seems now well organized and suitable for pubblication, so no further corrections are needed.

Best regards